# Exploring Lived Experiences of Adolescents Presenting with Self-Harm and Their Views about Suicide Prevention Strategies: A Qualitative Approach

**DOI:** 10.3390/ijerph18094694

**Published:** 2021-04-28

**Authors:** Anum Naz, Amna Naureen, Tayyeba Kiran, Muhammad Omair Husain, Ayesha Minhas, Bushra Razzaque, Sehrish Tofique, Nusrat Husain, Christine Furber, Nasim Chaudhry

**Affiliations:** 1Pakistan Institute of Living and Learning, Karachi 75600, Pakistan; anum.safdar90@gmail.com (A.N.); amna.naureen@pill.org.pk (A.N.); sehrish.tofique@pill.org.pk (S.T.); nasim.chaudhry@pill.org.pk (N.C.); 2Centre for Addiction & Mental Health, University of Toronto, Toronto, ON M5S, Canada; Omair.Husain@camh.ca; 3Institute of Psychiatry, Benazir Hospital, Rawalpindi 23000, Pakistan; ayeshaminhas@gmail.com (A.M.); bushrazzaque@gmail.com (B.R.); 4Division of Psychology and Mental Health, University of Manchester, Manchester M139PL, UK; nusrat.husain@manchester.ac.uk; 5Division of Nursing, Midwifery and Social Work, University of Manchester, Manchester M139PL, UK; christine.furber@manchester.ac.uk

**Keywords:** self-harm, suicide, adolescents, qualitative, Pakistan, framework analysis

## Abstract

Suicide is a serious global public health problem and the third leading cause of death in those 15–35 years old. Self-harm is the major predictor of future suicide attempts and completed suicide yet remains poorly understood. There is limited evidence on effective interventions for adolescents who present with self-harm. To identify and develop acceptable preventive and therapeutic interventions it is essential to understand the factors that contribute to self-harm and suicide in young people, in the context of their emotions, interpersonal difficulties, available support and prevention strategies. This qualitative study aimed at exploring the lived experiences of adolescents presenting with self-harm and their views about potential prevention strategies. Semi-structured interviews with 16 adolescents (12–18 years) presenting with a self-harm episode in a public hospital in Pakistan. A topic guide was developed to facilitate the interviews. The following themes emerged using the framework analysis; predisposing factors (interpersonal conflicts, emotional crisis etc.), regret and realization that self-harm is not the only option, perceived impact of self-harm, and suggestions for suicide prevention strategies (sharing, distraction techniques, involvement of family). This study may help in refining a contextual and culturally based explanatory model of self-harm in adolescents and in informing development of culturally acceptable interventions.

## 1. Introduction

Adolescent suicide is a serious public health concern worldwide [1]. According to the World Health Organization (2014) 800,000 people die due to suicide every year, which equates to one person every 40 s. Suicide occurs throughout the lifespan and is the second leading cause of death among 15–29 year olds globally [1]. Self-harm is one of the strongest predictors of death by suicide in adolescence [2,3].

Rates of self-harm in South Asia are particularly high compared to the global average [4]. However, these figures are likely to be an under estimate since self-harm and suicide data from many Low and Middle Income Countries (LMICs) is lacking and the available data is not reliable [4,5]. In India, the suicide rates were found to be much higher than the official estimates [6]: “suicide accounted for approximately as many deaths as transport accidents in males and maternal deaths for the age group of 15–29 years”. In a systematic review of mental health studies of adolescents in India three-month prevalence of self-harm was 3.9% to 25.4% in community based studies [7,8]. There is little official statistics on self-harm and suicide from Pakistan, a conservative Islamic country where suicide and self-harm are criminal acts, as well as socially and religiously condemned [5]. However, there is accumulating evidence that both self-harm and suicide rates are on a rise in Pakistan [5]. Self-harm also carries a huge economic burden because of large treatment costs associated with self-harm and suicide in LMICs such as Pakistan [9] and high income countries such as the United States [10] and the United Kingdom [11].

A clear understanding of risk factors is important for designing appropriate interventions with those at highest risk of suicide [12]. Psychiatric, psychosocial, and familial factors have been identified as significant contributors for self-harm [13] and suicide in adolescents [14]. Evidence from quantitative studies highlighted various risk factors of self-harm in adolescents; family conflict, truancy, school absenteeism, members of peer groups indulging in self-harm [13], and bullying [15,16]. Qualitative studies have shown that adolescents attribute several factors to their act of self-harm including negative emotions toward the self, lack of control over their lives, and disturbed interpersonal relationships [17,18].

Adolescents who self-harm are at elevated risk of repeating this behaviour later in life [2], therefore, intervention with this group may produce longer-term savings. Evidence is available on interventions that are recommended for young people who present with self-harm [19,20], however, many adolescents do not engage well with face-to-face services [21,22]. To improve adolescents’ access to and engagement with self-harm and suicide prevention strategies, there is a need to explore the adolescents’ lived experiences of self-harm and their views on what might prevent a self-harm episode [23]. Hence, exploring adolescents’ views may provide opportunities for developing culturally relevant and more acceptable suicide prevention strategies [23].

The aim of this qualitative study was to explore the perspective of adolescents who presented to a large public hospital in Pakistan with a recent history of self-harm (within last three months) in order to identify predisposing factors of self-harm, perceived consequences, reactions of family members and their needs for services including what an acceptable psychosocial intervention would be like.

## 2. Material and Method

### 2.1. Research Design

The qualitative research design was used to ascertain further explanations of adolescent’s experiences to provide information to inform the development of a suitable and acceptable intervention. This study used a pragmatic approach to explore experiences. Data were generated using face to face semi-structured interviews and analysed using framework analysis principles [24]. This approach is considered suitable in healthcare research to explore patients’ experiences in sensitive circumstances [25].

### 2.2. Setting

The study was conducted in medical units of Benazir Bhutto Hospital Rawalpindi (Pakistan). The setting for the interviews was chosen by mutual consensus between interviewers and interviewees and all interviews were conducted in a research office at Benazir Bhutto hospital. All participants were reimbursed for their time and travel.

### 2.3. Ethical Approval

Ethics approval was sought by the Institutionalized Review Board of Benazir Bhutto Hospital Rawalpindi Pakistan (ethics approval no# RWP/12/08/17). All researchers involved in the study were Good Clinical Practice (GCP) certified. Participants were approached and screened by the trained research assistants when they were admitted in the medical units after their self-harm attempt. After providing a description of the study, verbal consent was taken from the adolescents and their family for initial screening. Detailed participant information leaflet was provided to them and they were asked to discuss this with their families. Consent for further contact was taken from eligible participants and two contact numbers were taken. Participants were contacted again by the researchers through a phone call within two weeks of their discharge from hospital. It was made clear to the participants and their family that participation in the study was voluntary and they were informed by the interviewers that they are allowed to withdraw from the study at any time without giving a reason and without any detriment to themselves. Written consent was taken from all the participants as well as their parents. All interviews were audio—recorded with participants’ consent and these recordings were kept in password protected folders to ensure confidentiality and privacy. Identification numbers were assigned to all the participants and transcripts were anonymized. A distress policy was put in place and counselling and support were offered to the participants in case they felt distressed during or after the interview.

### 2.4. Sampling and Recruitment

Patients meeting the following inclusion/exclusion criteria were recruited.


**Inclusion criteria:**


In the context of this study self-harm was defined as, “an act with non-fatal outcome, in which an individual deliberately initiates a non-habitual behaviour that, without interventions from others, will cause self-harm, or deliberately ingests a substance in excess of the prescribed or generally recognized therapeutic dosage, and which is aimed at realizing changes which the subject desired via the actual or expected physical consequences.” [26]. Patients were eligible if they were;

Aged 12–18 years with a recent history of self-harm (within last three months)


**Exclusion criteria:**


Patients were excluded if they;

Had an ICD-I0 mental disorder diagnosis; due to a general medical condition or substance misuse, dementia, delirium, alcohol or drug dependence, schizophrenia, bipolar disorder, learning disability, that could prevent their participation in a qualitative interview.Were unable to engage, participate or respond to the interview questions.Needed inpatient psychiatric treatment because of severity of their psychiatric illness as this could prevent their participation in a qualitative interview.

### 2.5. Duration

The study was conducted over a period of 4 months (September–December 2017).

### 2.6. Data Collection

Semi-structured in-depth interviews [24] were conducted in Urdu by trained researchers (clinical psychologists who had previous experience of conducting qualitative interviews and were involved in a Randomized Controlled Trial of problem solving intervention for adults with history of self-harm—MR/N006062/1) after taking written consent from participants and parents. A semi structured topic guide to facilitate interviews was developed using the following steps;
A panel of experts including 5 experienced psychiatrists (one of whom was a child and adolescent psychiatrist) and 6 clinical psychologists with a previous and on-going experience of working with adults presenting with self-harm, had a 3-h long discussion session where they discussed and prioritised the possible areas to be added to the topic guide keeping in mind the purpose of the study and the use of appropriate words that would be understandable for the adolescents.At the second stage, two clinical psychologists and a senior psychiatrist developed a first draft of the topic guide using a detailed literature review of the topic and points made during the panel discussion.The first draft of topic guide was then sent to the child and adolescent psychiatrist to be evaluated for appropriateness of language and to ensure all relevant areas were included in topic guide. An English version of topic guide was also shared with an expert qualitative researcher for her feedback and suggested changes on question style to enhance the acquisition of in-depth data were incorporated.

The topic guide was comprised of open-ended topical questions followed by relevant probes to explore further information, and prompts to encourage further detail. The interviews were recorded with permission and transcribed verbatim and then translated into English to aid analysis. Each participant was interviewed on only one occasion. On average, interviews lasted for an hour. At the end of the interview, all participants were asked if they needed psychological support and three were offered initial counselling session and were referred for further support to the relevant services. Interviews were conducted till the data saturation was achieved. A total of 12 participants were approached to review the transcripts and suggest changes (however none of them asked for any changes).

Quality control and Assurance: Researchers underwent intensive training in qualitative data collection and analysis prior to the data collection. They attended a comprehensive training workshop on qualitative research and analysis in United Arab Emirates (UAE) that involved didactic presentations, development of topic guide, role play sessions to conduct interviews on sensitive issues such as self-harm, and hands on exercises using Framework analysis principles. A senior researcher reviewed all audio files and provided detailed feedback to the researchers to ensure high-quality interviews. Weekly supervision meetings were held to discuss study progress, provide additional training and supervision.

Risk Protocol: A risk assessment policy was also put in place to assess any risk of future suicide attempts and there was a protocol in place to refer such participants for psychiatric evaluation and treatment. No participants were found to be at risk during this qualitative study.

### 2.7. Data Analysis

Data were analysed using the five stages of a Framework Analysis, which involves familiarization of the data, identification of a theoretical framework, indexing, charting and mapping of themes and finally interpretation of themes [24]. In the first stage, all the transcripts were read several times by two qualitative researchers, listening to tapes and studying field notes allowed the researchers to immerse themselves and get familiar with the content of the interviews. Following detailed readings of the transcripts, a conceptual thematic framework was constructed to identify key issues, concepts and themes [27]. Development of the framework was based on three important components; a priori issues (informed by the topic guide), emergent issues raised by the participants themselves, and analytical themes emerging from the recurrence of particular views or experiences. Categories and subcategories were identified. The third step of the analysis was the indexing of themes, which included application of the draft thematic framework systematically to interview data in the textual form. Indexing references were written on margins of each transcript by descriptive textual system based on headings (categories and sub-categories) derived during development of thematic framework. Next, charting was performed. During the process of charting, data were lifted from the original context and rearranged and summarized in the form of table/charts. Charting enabled the researchers to develop a holistic sense of the data as a whole. The last step of the analysis involved mapping and interpretation. The purpose of this stage was to draw together key characteristics of the data, and to interpret the data set as a whole.

To maintain trustworthiness of the data and subsequent findings, the researchers in Pakistan were supervised by experienced qualitative researchers based in the UK and mental health experts in Pakistan. Regular fortnightly Skype meetings were held to discuss progress with the study during the data collection and analysis phases. Drafts of anonymised and password protected analyses were electronically mailed to the qualitative researcher in the UK who was able to study these before Skype meetings. The regular reviews by all researchers helped to ensure the fit of the data to the final analysis, and support minimization of bias [27]. Different team members including experts reviewed each stage of the analysis to minimise bias (Maher et al. 2018). Team members met to agree the final review of the theoretical framework. Furthermore, translations (Urdu–English) were back-translated to ensure accuracy. For respondent validation [28], a total of 7 participants were approached for feedback about the accuracy of the data they had given as well as the researchers’ interpretation of that data. A total of 5 were contactable. Researchers discussed main themes over the phone. This discussion took between 12 and 15 min. They acknowledged that the role of interpersonal problems is highlighted very appropriately, recommended to explain consequences of self-harm (psychosocial impact) and suggested that conveying this information to the readers/audience can help to stop them from self-harm.

## 3. Results

A total of 29 adolescents were approached and 22 of them met the eligibility criteria. Out of these eligible adolescents, 4 were not contactable after discharge from the hospital and the remaining 3 adolescents refused to participate in the study. A total of 16 adolescents participated in the study; 9 were female and 7 male. Their education varied between grade 9 (*n* = 1), grade 10 (*n* = 8), grade 11 (*n* = 5), and grade 12 (*n* = 2). All participants except one (drug overdose) used rat poison as a method of self-harm (Table 1).

Four themes emerged from the data, these were:

### 3.1. Predisposing Factors to Attempting Self-Harm

#### 3.1.1. Interpersonal Conflicts with Family

Participants described their environment as characterized by frequent interpersonal confrontations amongst family members.

“I don’t know why but there are a lot of arguments at my home all the time. Sometimes my father fights with my mother and sometimes my uncle.”(Participant 7)

“My younger brother created a lot of misunderstandings and told my father that I talk to boys over phone. My father did not give me a chance of clarification even once and started shouting. Nobody trusts me.”(Participant 2)

Participants gave a clear picture of their conflicting relationships with their parents particularly in terms of how parents discriminate among siblings and do not take care of participants’ self-respect.

“Whatever happens, my mother supports my brother, even when he (brother) slaps me.”(Participant 5)

“My mother insulted me in front of everyone.”(Participant 15)

#### 3.1.2. Emotional Crisis

Participants reported emotional crisis that they were experiencing before the act of self-harm including fear, anger, aggression, feeling tensed and helpless.

“I was so fearful.”(Participant 10)

“I feel angry when everybody at home calls me drug addict just because I smoke (cigarettes).”(Participant 13)

“Next day when I woke up I was feeling very tense over the fight I had.”(Participant 4)

“I was feeling helpless I wanted to get rid of them (family)”.(Participant 4)

Participants also reported feeling alone and isolated

“I have no one to talk to. Nobody listens.”(Participant 6)

“I was feeling all alone.”(Participant 5)

#### 3.1.3. Finding Self-Harm as an Only Option

During the interviews all participants gave detailed description of their disturbed state of mind, and the thoughts that led to the self-harm act.

“I was fed up with tough and busy life and I had a constant feeling that nothing good will happen to me.”(Participant 5)

“Scenes from my past were going on in my mind like a film and I was thinking nothing will change in my life.”(Participant 7)

Self-harm was closely linked to a situation with which the adolescent could not deal —all efforts were perceived to be in vain. Self-harm thus became the only possible way to send a message that was impossible to deliver otherwise.

“I thought I am the only daughter but they (parents) need a son, if I live what can I do for them? I am better off dead.”(Participant 9)

“My parents were angry with me, and were trying to make me tense; I thought today is the day and I will kill myself.”(Participant 4)

They expressed their act of self-harm as a way to end intolerable conditions and they perceived that self-harm would help them to get over the difficult life situation.

“I will get rid of all of them.”(Participant 6)

“I wanted to end the discrimination that I was experiencing at my home by ending my life.”(Participant 16)

### 3.2. Reflections after Self-Harm: Regret and Realization That Self-Harm Is Not the Answer

Participants expressed that they themselves were responsible for their act of self-harm and their family had to face negative reaction from the society because of their act and that made them feel bad.

“No, it (self-harm) was my mistake and I won’t do this again.”(Participant 1)

“No one is responsible. I myself am responsible (for act of self-harm).”(Participant 4)

“I feel bad because I don’t blame my parents for any of it yet they have to face the entire backlash for no reason and it makes me feel bad.”(Participant 14)

They realized that one should be thankful for whatever they have in their lives and regretted the self-harm act.

“A person should value and appreciate small things in life.”(Participant 9)

“I regretted this act (of self-harm) afterwards.”(Participant 15)

### 3.3. Perceived Impact of Self-Harm

Psychosocial Impact:

Narratives related to the post self-harm period highlighted the impact of their act on family and on themselves. Participants shared that family members were worried, embarrassed, and angry over their act.

“My parents are also embarrassed because of my act (of self-harm).”(Participant 9)

“My parents were extremely worried, father was angry and he said why you took such a big step for such a minor issue.”(Participant 1)

“They don’t allow me to go to school after the incident.”(Participant 2)

“I was the reason of her (grandmother) heart attack.”(Participant 4)

Participants emphasized worsening of certain situations after their self-harm such as feeling more upset, depressed and embarrassed.

“I can’t study, the whole environment is disturbed now, and that’s why I feel even more stressed and tense.”(Participant 7)

“I had to quit my job; I cannot face anyone at my work place.”(Participant 4)

“Bad reputation and insult these are two things I am left with.”(Participant 15)

However, some participants reported feeling less miserable as well and they believed that their self-harm brought a positive change in their family’s behaviour.

“It is less than the misery that I used to feel every day.”(Participant 10)

“My parents realized their mistake I believe their behaviour is quite better with me now.”(Participant 8)

### 3.4. Suggestions for Prevention Strategies

Participants came up with some very useful suggestions regarding type of help that they think can benefit adolescents. They emphasized on the role of emotional ventilation and having a confidant.

“Taking advice from others may prevent it (self-harm).”(Participant 7)

“There should be someone who can ask them about their problem.”(Participant 4)

“I think discussion is important, those who do not do this suffocate themselves from inside, and sharing can help to relax.” (9)

They also talked about the idea of distraction technique and behavioural activation in the form of activity scheduling as well as the role of learning better ways to solve problems.

“At times it helps to distract ourselves.”(Participant 4)

“By keeping oneself busy all the time, there should be somebody for their (adolescents at risk of self-harm) supervision so that they would not feel bad and we should try to keep them happy.”(Participant 9)

“Helping people to learn how to make good decisions can help them to think about different alternatives and to stay away from such acts (of self-harm).”(Participant 6)

Participants highlighted that the adolescents can be encouraged and motivated through use of stories from lives of famous people/celebrities.

“Stories should be told to young children because they get influenced by the stories. The hero in the story is their role model and they get motivated by what he does.”(Participant 8)

“There should be stories of those who wanted to commit suicide but then found a better way, stories with some message or moral.”(Participant 9)

“Show them videos… why people commit suicide and talk about these stories.”(Participant 5)

“Sharing stories of people who later realized that their act of self-harm was not right, would be helpful.”(Participant 8)

Participants particularly talked about the role of involving family members in any intervention being offered to the self-harm survivors.

“Involving family and friends will help because we listen to them.”(Participant 5)

“Families should be involved so that they learn that it is important to give proper time (to the adolescents).”(Participant 7)

“Families always have good intentions for us they can be involved to give advice.”(Participant 9)

## 4. Discussion

This is the first qualitative study from Pakistan to investigate the adolescents’ lived experiences of self-harm behaviour and to explore their views on how a potential intervention to prevent self-harm should look like. We found that adolescents who self-harm are deeply affected with their attempts and came up with helpful suggestions for developing interventions as part of prevention strategies and management plan for Pakistani adolescents.

Results highlighted that all participants except one (drug overdose) used rat poison as a method of self-harm. This is consistent with previous evidence from Pakistan which show that the use of insecticides and pesticides are common methods of self-harm in this population [5,29,30]. These findings have important implications for suicide prevention as the World Health Organization (WHO) has recommended to limit the access to the means of suicide as an effective method of preventing suicide [1].

Our findings from analysis of qualitative interviews are consistent with previous work. The emotional crisis, one of the predisposing factor reported by the participants, show the subjective experiences of negative emotions towards the self and feelings of loneliness. The experience of self-harm act described by the adolescents is primarily a solitary experience involving negative emotions such as anger, fear, sadness and helplessness. Negative emotional experiences are consistently demonstrated in previous studies with suicidal adolescents [31,32].

Similarly, interpersonal conflicts with family members were reported as a predisposing factor. There is already established evidence on a number of family and social factors which increase the likelihood of self-harm behaviour in adolescents such as maladaptive parenting, stress related to parenting, domestic violence, lack of support and child maltreatment [33,34]. Gender discrimination by the parents was reported as a potential triggering factor by the female adolescents (such as giving preference to son over daughter and supporting their acts), which is quite common within Pakistani culture, as Pakistan is a male dominant society, where males have more authority and power and females are treated as subordinates [35]. Gender discrimination by the parents plays an important role in developing an inferiority complex and is marked as a mediator/triggering factor of poor mental health conditions [36]. In this current study participants perceived self-harm as an only option. This was seen as their main drive to self-harm—to get relief from negative thoughts and emotions, and to get help. These findings are similar to those reported in published studies on self-harm in young people [18,23]. Participants reported feelings of regret and realized that self-harm was not the only option when they had a chance to reflect on their self-harm attempt. Several studies quoted that not only adolescents but their parents also felt regret and shame after self-harm attempt and these reactions were mainly linked with the fear or stigma attached to self-harm and suicide behaviours [37], particularly in the context of Pakistan where both self-harm and suicide are illegal acts and are socially and religiously condemned [5]. These negative experiences not only increase parental burden, distress, disappointment and social isolation but also affect parent-child relationship and overall family atmosphere/environment [37]. In addition, participants’ feelings of regret can also be explained in cultural context. Pakistani families are mostly collectivistic, patriarchal and large where the elder family members make the major decisions for all family members [38]. Since the childhood, family members are encouraged to conform to the norms and rules of the family and society [39]. Therefore, adolescents may feel obliged to express their feelings of regret over self-harm act to protect repute of the family considering the ingrained notion of *izzat* (honour) in the Asian families [40]. All these psychosocial factors endorsed a great need for development of culturally relevant support programs for the adolescents and families in order to understand feelings and behaviours that trigger and maintain self-harm.

Our findings highlight the importance of the potential role of service users and experts by experience in developing culturally relevant prevention strategies. Evidence from a previous qualitative study showed that adolescents want to actively participate in mental health research process and they emphasized that meaningful involvement of adolescents in research can play a major role in ensuring relevance of research projects [41]. In the current study, adolescents highlighted the importance of emotional catharsis and adopting a healthy routine and activities as part of distraction technique and to learn problem solving in order to cope well. An exploratory study has evaluated the role of problem solving intervention which is based on the principles of the Cognitive Behaviour Therapy (CBT) for suicide prevention in Pakistan [30]. However, for young people this evidence is largely limited to high income countries only [20] and lots more work still needs to be done in LMICs such as Pakistan. Moreover, participants discussed the need and importance of involving family members in potential interventions designed for self-harm and suicide prevention. Empowering parents in the intervention programs play a vital role in reducing risk of future self-harm and improve help-seeking behaviour as well [42,43].

## 5. Strengths and Limitations

This was the first qualitative study from Pakistan exploring adolescents’ perspective on self-harm and their views on what potential self-harm and suicide prevention strategies should look like. Regular supervision from national and international academics and mental health professionals contributed to reducing the risk of bias and misinterpretation of the data, hence increasing the trustworthiness of the findings. As the participants in this study were recruited from only one hospital in one city in Pakistan, the perspective of adolescents in other regions may differ. Furthermore, these findings may not be generalizable to other areas, particularly rural areas. Additionally, it is also important to understand caregivers’ and health professionals’ perspectives about adolescent self-harm and suicide.

## 6. Conclusions

This qualitative study is an important contribution towards developing a contextually and culturally based explanatory model of self-harm in adolescents. Findings from this study support the major role of qualitative research in self-harm and suicide prevention research. Use of direct verbatim in qualitative research illuminates how a variety of predisposing factors create a pathway to self-harm. The emergent nature of qualitative enquiry not only allows for the generation of potential themes from research participants themselves, but also provides an inductive approach to developing self-harm and suicide prevention approaches.

## Figures and Tables

**Table 1 ijerph-18-04694-t001:** Sample Characteristics.

S.no	Age	Gender	Education	Method
1	16	Female	Grade 10	Rat poison
2	17	Female	Grade 11	Rat poison
3	16	Female	Grade 12	Rat poison
4	16	Female	Grade 9	Rat poison
5	17	Female	Grade 10	Rat poison
6	17	Male	Grade 11	Rat poison
7	16	Female	Grade 11	Rat poison
8	16	Female	Grade 10	Rat poison
9	17	Male	Grade 11	Rat poison
10	17	Female	Grade 10	Rat poison
11	17	Male	Grade 10	Rat poison
12	17	Male	Grade 10	Rat poison
13	14	Male	Grade 10	Medicine Overdose
14	15	Female	Grade 11	Rat poison
15	17	Male	Grade 10	Rat poison
16	17	Male	Grade 12	Rat poison

## Data Availability

Data (in the form of anonymised transcripts) will be available on request.

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
