# Peer review of "Exploring Lived Experiences of Adolescents Presenting with Self-Harm and Their Views about Suicide Prevention Strategies: A Qualitative Approach"

_ijerph, 2021, doi:10.3390/ijerph18094694_

Round 1
Reviewer 1 Report
This research article describes a qualitative study exploring the perspectives of 17 Pakistani adolescents with a history of recent self-harm who presented for inpatient treatment at a major hospital center. The goal is to better understand the lived experiences of these young people, which could inform development of suicide prevention strategies for this population. The authors conducted clinical interviews with participants and used a framework analysis to identify core themes (e.g., predisposing factors, perceived impact) which they then use to recommend possible suicide prevention strategies and areas for future research/intervention.
The manuscript addresses an important topic of broad public health interest. A key strength is its inclusion of a population that has received little research attention in terms of suicide risk/prevention (Pakistani/South Asian adolescents) yet may be at high risk for suicide. The manuscript is generally well-written, aside from a few minor English language issues that are easily fixed. Based on the study results, the authors make useful recommendations for developing culturally-acceptable suicide prevention and intervention programs, and for future research. There are a few areas for improvement, including parts of the methods section that are a bit underdeveloped, and some other areas that could use minor clarifications or expanded discussion. With minor revisions to address these issues, the article should make a useful addition to the literature.
Specific comments
Abstract:
- Line 12: Should be “in those 15-35 years old”.
- Line 16: Should be “it is”
- Line 19: Should be “Semi-structured”
- Line 21: Should be “The following themes”
Introduction:
- I’m not sure what is meant by “associated with large treatment costs in Pakistan and other contexts” – is this a typo?
- The authors seem to use the terms “self-harm” and “suicidality” interchangeably in this section, but the risk for suicide is likely different between adolescents with a history of prior suicide attempts and those with only non-suicidal self-injury. Is there any data about how the nature of past self-injurious behavior relates to suicide risk in this population? Discussing this in more depth could be helpful.
- Line 60: Should be “what an acceptable psychosocial intervention would be like”
Methods:
- Under “Research design”, it would help to be more specific about the specific qualitative methods chosen, and why
- The inclusion criteria are confusing to me. It seems that participants were screened out if they needed psychiatric inpatient admission, but wouldn’t that limit enrollment to only those with relatively minor self-injury, or without suicidality? It would help to expand and clarify these criteria and the rationale
- Inclusion criterion #2 seems like it is really an exclusion criterion
Results/Discussion:
- I was surprised that all participants except for one had self-harmed by ingesting rat poison. Is this method common in Pakistan? Although it is sometimes used in Western countries, I believe it is less common than medication overdoses or other methods. I would be interested to hear what the authors make of this finding.
- I also wonder what the authors think about the fact that most participants seemed to express regret about having self-harmed. As they note in the introduction, self-harm is a crime in Pakistan, and given societal/religious condemnation of self-injurious, I would imagine that adolescents experience a great deal of pressure to denounce it. However, there could be other interpretations too. This could make an interesting addition to the discussion.
- Line 332: This sentence is confusing to me – could it be reworded?
- Line 334: Should be “lots more work”
Reviewer 2 Report
Very interesting topic. The authors do not mention anything about bullying, is it not an issue in Pakistan? It would be interesting if the discussion addressed it.
Reviewer 3 Report
This manuscript presents a relevant topic to publish in IJERPH, which could be accepted with some major revision. The manuscript must have a profound linguistic, grammatical and spelling revision before its publication.
In my opinion, the introduction provides adequate information and structure to set up the research questions raised in the manuscript; the methodology provides sufficient detail; the results section is sufficiently clear and precise; the discussion of results based on previous literature.
After carefully reading your manuscript, I point out some aspects that must be improved and corrected:
- Some aspects of formatting and should be corrected (spelling, punctuation). Please, correct what is pointed out in the body of the manuscript;
- All statistical symbols must be in italics (n ).
- The authors use abbreviations throughout the manuscript that are not intuitive and sometimes not even necessary. See some comments I made throughout the manuscript.
Reviewer 4 Report
Deficiencies in bibliographic updating.
They use outdated references to refer to epidemiological data.
Important methodological deficiencies: biases in the selection of the sample or reliability of the interviews used.
They use ICD-I0 and not ICD-11.
The information collected (results) is more than three years old.
The study does not provide generalisable data.
Round 2
Reviewer 4 Report
Methodological deficiencies persist.
They have not responded to all the questions raised in the review.
Author Response
Comment: Methodological deficiencies persist.
They have not responded to all the questions raised in the review.
Response:
We are thankful to the reviewer for reviewing the revised manuscript. We have looked at the methodology section again and tried to add some further details such as trustworthiness of the data and findings. However, it would be useful if we can have specific comments from reviewer 4 so we can make further changes.